# A point-of-care microfluidic biochip for quantification of CD64 expression from whole blood for sepsis stratification

U. Hassan[1,2,3], T. Ghonge[1,3], B. Reddy Jr.[2,3,†], M. Patel[1,3], M. Rappleye[1,3], I. Taneja[1,3,†], A. Tanna[1,3,†], R. Healey[1,3], N. Manusry[1,3], Z. Price[1,3], T. Jensen[3], J. Berger[1,3], A. Hasnain[1,3], E. Flaugher[1,3], S. Liu[1,3], B. Davis[3], J. Kumar[3], K. White[3] & R. Bashir[1,2,3,4]

Sepsis, a potentially life-threatening complication of an infection, has the highest burden of death and medical expenses in hospitals worldwide. Leukocyte count and CD64 expression on neutrophils (nCD64) are known to correlate strongly with improved sensitivity and specificity of sepsis diagnosis at its onset. A major challenge is the lack of a rapid and accurate point-of-care (PoC) device that can perform these measurements from a minute blood sample. Here, we report a PoC microfluidic biochip to enumerate leukocytes and quantify nCD64 levels from 10 μl of whole blood without any manual processing. Biochip measurements have shown excellent correlation with the results from flow cytometer. In clinical studies, we have used PoC biochip to monitor leukocyte counts and nCD64 levels from patients' blood at different times of their stay in the hospital. Furthermore, we have shown the biochip's utility for improved sepsis diagnosis by combining these measurements with electronic medical record (EMR).

[1] Department of Bioengineering, University of Illinois at Urbana-Champaign, 1270 Digital Computer Laboratory, 1304 W. Springfield Ave., Urbana, Illinois 61801, USA. [2] Micro and Nanotechnology Lab, University of Illinois at Urbana-Champaign, 208 N. Wright St., Urbana, Illinois 61801, USA. [3] Biomedical Research Center, Carle Foundation Hospital, 509 W University Ave., Urbana, Illinois 61801, USA. [4] Carle Illinois College of Medicine, 807 South Wright St., Urbana, Illinois 61801, USA. † Present address: Prenosis Inc., 210 Hazelwood Drive, Suite 103, Champaign, Illinois 61822, USA. Correspondence and requests for materials should be addressed to R.B. (email: rbashir@illinois.edu).

Intensive care units (ICUs) in the United States receive more than 5 million patients annually[1,2]. Of these, severe sepsis strikes more than 1 million people, or roughly 20% of all ICU patients, with an overall cost of about $24 billion to the healthcare system[2]. An estimated 28–50% of these people die (280,000–500,000), a number which is greater than the number of American deaths from prostate cancer, breast cancer and AIDS combined[3]. A dominant factor underlying these grim numbers is the lack of an accurate, rapid sepsis diagnostic methods at the point of care (PoC)[4]. The current standard for diagnosis, termed as systemic inflammatory response syndrome (SIRS) criteria, is monitoring increased temperature, respiratory rate, $PaCO_2$ levels in blood and abnormal total WBC count. This is followed by a 1–3 day bacterial growth culture for the pathogen, followed by nucleic acid identification. The diagnostic process takes longer than the disease progression, thus leaving huge diagnostic gaps in the treatment pathway[5]. Several promising biomarkers based on inflammatory response have been reported, most notably neutrophil cluster of differentiation (CD64)-positive cells, which is a high-affinity biomarker that binds to immunoglobulin G. It is normally expressed on monocytes but in cases of inflammation, CD64 expression is upregulated rapidly on neutrophils[6–8]. During infection or inflammation, an increase in the expression of CD64 on neutrophils is stimulated by inflammatory cytokines. The intensity of the cytokine stimulus is directly correlated with the graded increase in the CD64 expression[6,7]. Many meta-analysis studies have shown that, particularly when SIRS is combined together with neutrophils' CD64 + cells, the accuracy, sensitivity and specificity of sepsis diagnosis at early stages of disease can be dramatically improved[8–10]. Diagnosing sepsis early is extremely critical, as several innovative treatment strategies are now available that can markedly increase chances of survival if applied early enough in the appropriate situations, including antimicrobial therapies, and immune-stimulating and immunosuppressive therapies[11–13]. The 72-h survival rate decreases by roughly 7.7% every hour such that appropriate antimicrobial medication is delayed at the onset of infection, underscoring the need for early diagnosis techniques[13].

Currently, haematology analyzers are being used for complete blood cell counts and flow cytometers for specific leukocyte counting. Antigen expression-based cell quantification is mainly performed by a flow cytometer. However, these instruments have yet to find widespread use in the PoC settings due to several reasons. First, flow cytometry measurement suffers from a lack of standardization. Reproducible protocols for sample preparation including RBC lysis, cell staining, gating strategies and acquisition protocols have proven difficult to be kept constant for multicentre clinical studies[14]. Second, flow cytometry measurement and haematology tests require both a well-equipped laboratory and significant technical expertise, which are impossible to maintain 24/7 in ICUs.

Much effort has been placed in creating PoC microfluidic biochip solutions for clinical diagnosis. Towards specific leukocyte enumeration, many efforts have used fluorescent tagging and subsequent image processing to automatically enumerate specific leukocytes in microchambers[15]. Some designs relied on the even distribution of cells in a plastic chamber to produce accurate counts[16]. Others have used a microfabricated membrane to filter out erythrocytes and recover the leukocytes, which were then fluorescently labelled[17]. Cheng et al. have investigated CD4 + T-cell capture by controlling shear stresses at the chamber walls and enumerating cells using a cocktail of fluorescently labelled antibodies[18,19]. They improved their design by including a monocyte depletion chamber to reduce the positive bias created at lower CD4 + T-cell concentrations[20]. The aforementioned optical methods require

the use of lenses and focusing to analyse samples, but this can increase the cost and decrease the portability of the device. Wang et al. further simplified the optics by not requiring an external light source: immobilized cells were labelled with CD3-conjugated horseradish peroxidase to facilitate a chemiluminescent reaction, which was amplified and quantified by a photodetector[21]. An electrical PoC solution would require only solid-state components to electrically interrogate a sensing geometry, process sensor output and provide input from and results to the user. Recent advancements have used the Coulter principle to electrically analyse individual cells within a population[22]. Adams et al. enumerated low concentrations of circulating tumour cells in whole blood samples after specifically capturing and releasing those cells[23]. Holmes et al. have used impedance analysis at multiple frequencies as a label-free method to differentiate between different leukocyte subsets, and were able to further enhance electrical differentiation by specifically attaching latex beads to the desired cells[24,25]. Our group has previously developed a microfluidic biochip capable of performing blood cell counts from a drop of whole blood without any manual processing[26]. The electrical sensors are designed to count individual cells using the coulter principle, where the passage of a cell perturbs the electrical current within an orifice, creating a distinct impedance pulse[26–28]. The biochip has shown good correlations in clinical studies when its cell counts were compared to the cell counts from haematology analyzers. We have also developed a microfabricated AC impedance analysis system for the electrical counting of specific leukocytes; in particular, we have enumerated CD4 + and CD8 + T cells in a microfluidic biochip by selective lymphocyte capture based on immunoaffinity and differential lymphocyte counting[29,30].

To date, very little work has been done to develop microfluidic devices to quantify the cell surface antigen expression levels. Murthy et al. reported tandem, spiral-shaped microfluidic devices to separate human umbilical vein endothelial cells and human microvascular endothelial cells based on different expressions of common antigen CD31 (ref. 31). They were able to achieve high purity of 80% for human umbilical vein endothelial cells and 97% for human microvascular endothelial cells, but did not validate their study with actual blood samples[31]. More recently, Pappas et al. have also developed microfluidic chips with a herringbone structure, and they have shown that the capture ratio of Ramos B lymphocytes and HuT 78 T lymphocytes matched the expression ratio of CD71 for the two cell lines after spiking in the blood samples[32]. However, to date there have been no reports of quantifying the expression level of antigens at the point of care on specific leukocytes like CD64 neutrophils from a minute volume of whole blood. None of the above-mentioned studies have the capability to perform PoC analysis from whole blood without the need for off-chip sample processing to quantify the expression level of the antigen on the cell's surface.

In this study, we have used our differential immunoaffinity capture technology to electrically quantify antigen expression level on the CD64 + cells. In particular, we have quantified CD64 expression levels on granulocytes + monocytes population by their selective capture in the biochip. We have performed clinical studies of the biochip at Carle Foundation Hospital, Urbana, IL, using blood samples collected from the patients who are SIRS-positive and/or their blood culture is ordered by the physicians at the time of their admission.

## Results

**Clinical studies of electrical cell counting from biochip.** We have developed an experimental assay to quantify the expression level of CD64 antigen on the neutrophil's membrane surface

based on the capture of CD64+ cells using our differential cell-counting immunocapture technology. The overview of the experimental assay for the differential expression-based cell-counting technology is shown in Fig. 1a. The image of the biochip with individual interconnected modules is shown in Supplementary Fig. 1. Whole blood (10 μl) is infused into the biochip at inlet 'a' along with lysing buffer at inlet 'b', to preferentially lyse erythrocytes (Supplementary Fig. 2). Formic acid in the lysing buffer creates the hypotonic extracellular environment for effective erythrocyte lysis, while saponin in the lysing buffer further helps to dissociate the erythrocyte debris clumps. The lysing process is halted by infusing quenching solution at inlet 'c' as the isotonic extracellular environment is achieved. Cells are electrically counted based on their size using microfabricated electrodes at the entrance counter. The design of the electrical counter with top and side views is shown in Supplementary Fig. 3. The anti-CD64 (clone 10.1) antibody is initially adsorbed in the cell capture chamber (Immobilization protocol in Methods section). The CD64+ cells are captured based on the CD64 expression level on their surfaces. The remaining cells are counted once more by the second electrical counter. The difference in the cell counts linearly correlates with the CD64 expression level on the cells. Electrodes are fed with 303 kHz and 5 V signal, as the cells flow through the counting aperture of 30 μm × 15 μm, and the resulting increase in the impedance is translated to voltage using a simple bridge circuit (Supplementary Fig. 4). A voltage pulse is produced with each passage of the cell through the counting aperture, with pulse amplitude representing the size of the cell. The cell pulses data are further digitally filtered to remove any unwanted noise and baseline drifts (Supplementary Fig. 5). Lymphocytes (6–10 μm) being smaller in size produce smaller pulses as compared to granulocytes/monocytes (11–15 μm) (Supplementary Fig. 6). The resulting histogram of peak pulse voltages is shown in Fig. 1b, showing lymphocytes and granulocytes + monocytes as two

distinct populations. The pulse area histogram obtained by considering pulse amplitude and pulse width is shown in Supplementary Fig. 7. As can be noted, the lymphocytes and granulocytes + monocytes populations are not distinct. Especially, the lymphocytes' population is merging with the noise, which can be attributed to much smaller pulse width of lymphocytes as compared to that of granulocytes + monocytes. We varied the flow rates from 10 to 30 μl min$^{-1}$ and measured the baseline noise (three times of the standard deviation of noise when no cell is flowing) of the cell-counting system, which increases with increased flow rates (Supplementary Fig. 8a). Furthermore, we also measured the pulse width of lymphocytes and granulocytes + monocytes population (Supplementary Fig. 8b). The pulse width at higher flow rates is smaller and also has higher standard deviation, making corresponding cell populations hard to distinguish in area histograms. In clinical studies, we measured biochip total leukocyte count and compared it to control leukocyte count obtained from a haematology analyser at Carle Foundation Hospital, Urbana. We found a correlation coefficient $R^2 = 0.89$ for 181 patient blood samples (Fig. 1c). We also compared the total granulocytes + monocytes counts from biochip versus control counts obtained from a haematology analyzer and found a high correlation coefficient of ($R^2 = 0.88$) (Fig. 1d).

**Clinical validation of nCD64 for sepsis stratification**. We also developed a CD64 expression quantification protocol on a flow cytometer to be used as control against our biochip (Methods Section). The stepwise protocol and flow cytometry analysis is shown in Supplementary Fig. 9. Supplementary Table 1 shows the dynamic range, average and standard deviation values of key parameters (leukocytes, lymphocytes and granulocytes + monocytes) on our targeted patient population. We have measured nCD64 expression values for $n = 450$ patient blood samples. The CD64 expression histograms for monocytes,

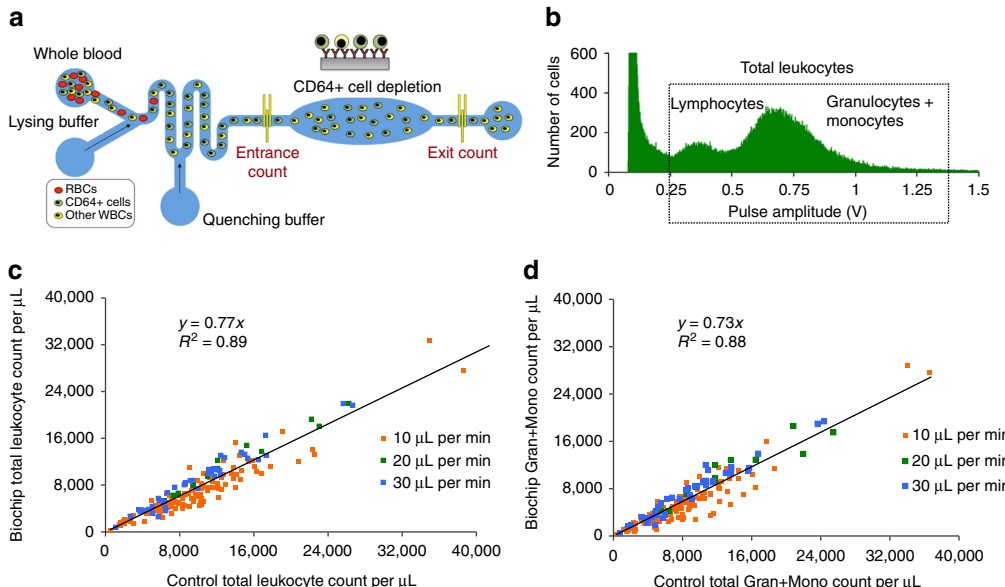

**Figure 1 | Overview of the electrical cell-counting technology.** (**a**) Process schematic of the differential expression-based cell-counting technology. Whole blood (10 μl) is infused in the biochip along with lysing and quenching buffers, to preferentially lyse erythrocytes. Cells are electrically counted and differentiated based on their size using microfabricated electrodes. Anti-CD64 (clone 10.1) antibody is initially adsorbed in the chamber. The CD64+ cells get captured based on their CD64 expression level. The difference in the cell counts from cell counters is linearly correlated with the nCD64 expression level. (**b**) The resulting pulse amplitude histogram representing lymphocytes and granulocytes + monocytes as two distinct populations. (**c**) Correlation (coefficient of determination: $R^2 = 0.89$, $P < 0.0001$) in between biochip total leukocytes versus control leukocyte counts obtained from haematology analyzer using $n = 181$ blood samples. (**d**) The correlation (coefficient of determination: $R^2 = 0.88$, $P < 0.0001$) in between biochip total granulocytes + monocytes versus control cell counts obtained from haematology analyzer.

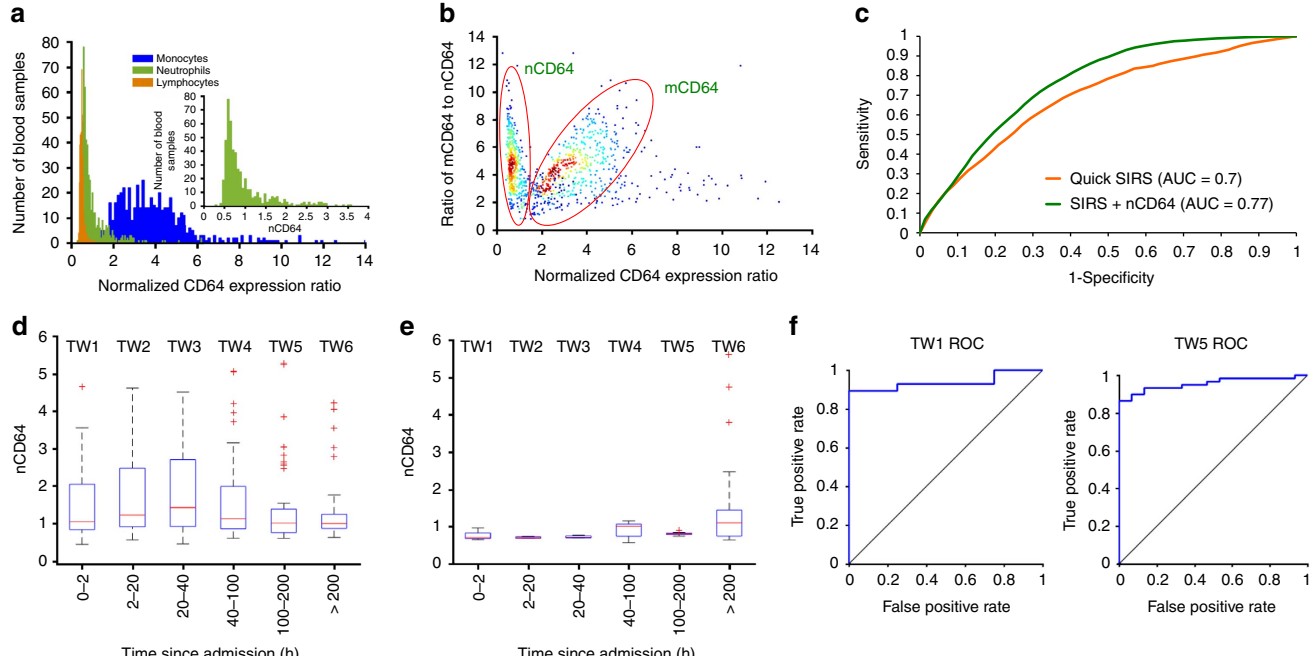

**Figure 2 | Clinical validation of nCD64 for sepsis stratification and prognostication.** (**a**) The CD64 expression histograms for monocytes, neutrophils and lymphocytes for $n = 450$ blood samples. The inset histogram represents nCD64 expression for all the blood samples. (**b**) The plot represents the ratio of mCD64/ nCD64 versus nCD64. (**c**) ROC curves showing sepsis predictability using Quick SIRS (AUC = 0.7) and SIRS + CD64 (AUC = 0.77). (**d**) The box plots showing the control nCD64 expression value from 316 blood samples collected from 68 patients (who recovered later) at different time windows of their hospital stay. (**e**) The box plots showing the control nCD64 expression value from 94 blood samples collected from six patients who are at different times of their hospital stay. Unfortunately, these patients did not recover. (**f**) The nCD64 and WBC counts are used for sepsis prognosis for all the six time windows using ANN model. ROC curves for TW1 (left) and TW5 (right) show the highest AUC > 0.9 for sepsis prognosis by predicting the recovery of the patients. P-values are reported in Methods section.

neutrophils and lymphocytes are shown in Fig. 2a. The inset histogram represents neutrophil CD64 expression. The increase in nCD64 is a result of neutrophil activation during proinflammatory response from an infection. We also investigated the increase in the monocyte CD64 (mCD64) expression comparable to neutrophil CD64 expression. The plot shown in Fig. 2b represents the ratio of mCD64:nCD64 versus nCD64. Increase in the monocyte CD64 expression as compared to nCD64 increases linearly with the increase in mCD64 expression. However, the increase in the nCD64 expression as compared to mCD64 expression is uncorrelated with the increase in the nCD64 expression in disease states.

We used the parameters to be measured by the biochip combined with electronic medical record (EMR) data as input features into a support vector machine (SVM) model to predict the onset of sepsis for our patient population (septic = 76 and non-septic = 368). Supplementary Table 2 shows the patient characteristics including age, gender, chronic conditions and common infections with respect to septic and non-septic cohorts in our patient population. Figure 2c shows receiver operating curves (ROC) with area under curve (AUC) for predicting sepsis using a subset of the SIRS criteria parameters (AUC = 0.70), and all the SIRS criteria parameters plus lactic acid combined with biochip parameters (AUC = 0.77). The subset of SIRS criteria parameters chosen (which we refer to as 'Quick SIRS') includes the SIRS parameters whose measurements can be outputted instantaneously. When the biochip measureable parameters are combined with the Quick SIRS criteria and lactic acid, the AUC increases from 0.70 to 0.77. To assess the importance of the SIRS criteria and biochip parameters, we report the weight vector coefficients outputted by SVM. The resulting output coefficients by the model are given in Supplementary Fig. 10.

Granulocytes + monocytes, white blood cell count, nCD64 and pulse were determined to be the four most important parameters in descending order. As three of these can be determined by our biochip, these results indicate that the biochip enables significant predictive power as compared to only measuring the Quick SIRS parameters (AUC = 0.70) in a short timeframe. However, in this same timeframe, the biochip will allow a clinician to measure other parameters (nCD64 and blood cell counts) that can improve the accuracy in diagnosing sepsis (AUC = 0.77). Since diagnosing sepsis is time-sensitive, a device quickly providing the measurements of parameters would be very useful.

In a second study, using the data collected on the patients, we also investigated the role of nCD64 and total leukocyte counts in septic patients' prognosis, in particular, predicting their outcome, that is, recovery versus non-recovery. Figure 2d shows the box plots for the nCD64 expression value from 316 blood samples collected from 68 patients at different times of their hospital stay. All of these patients recovered as also indicated by the increase and then subsequent decrease in the nCD64 expression value. Figure 2e shows the box plots for the control nCD64 expression value from 94 blood samples collected from six patients at different times of their hospital stay. Unfortunately, these patients did not recover and lost their lives. Their nCD64 value again increased from TW5 to TW6. We think that nCD64 can be an indicative biomarker for the recovery outcome, thereby emphasizing continuous nCD64 monitoring at different times throughout the individual patient hospital stay. Supplementary Figs 11 and 12 show the box plots for total leukocytes' counts and normalized ratio of nCD64 to WBC for both recovered and non-recovered patients. To investigate which time windows have high predictability for the sepsis prognosis, we developed an artificial neural network (ANN) model to predict the recovery of

patients using nCD64 and total leukocyte counts as input parameters. The nCD64 and WBC counts are used for patient prognosis for all the six time windows (Supplementary Fig. 13). ROC curves shown in Fig. 2f for TW1 (left) and TW5 (right) show the highest AUC>0.9 for predicting the recovery of the patients. The ROC curves for TW2, TW3, TW4 and TW6 are shown in Supplementary Fig. 14. The corresponding Hinton diagrams for all the time windows are shown in Supplementary Fig. 15. The weight and bias values for all time windows are given in Supplementary Table 3.

**Non-specific cell capture in a blocked chamber.** We designed the capture chamber to ensure that the cells experience maximum time interacting with the pillars, where antibodies are adsorbed and different-sized cells should experience the same amount of interaction time in the chamber. Furthermore, cell capture in the zero-velocity or stagnation regions around the pillars is minimum (Design details in Methods section). The first step in the expression-based cell capture investigation is to ensure the minimal non-specific cell capture in the blocked chambers with no immobilized antibody. To investigate this, we blocked the chambers with PBS + 1% BSA (protocol in Methods section) and find the cells captured by labelling them with fluorescent antibodies and counting by flow cytometer. Supplementary Fig. 16 shows the non-specific capture of cells in a blocked chamber. In Supplementary Fig. 16a, using a representative high CD64 expressing blood sample 'A' (with mCD64 = 6.03,

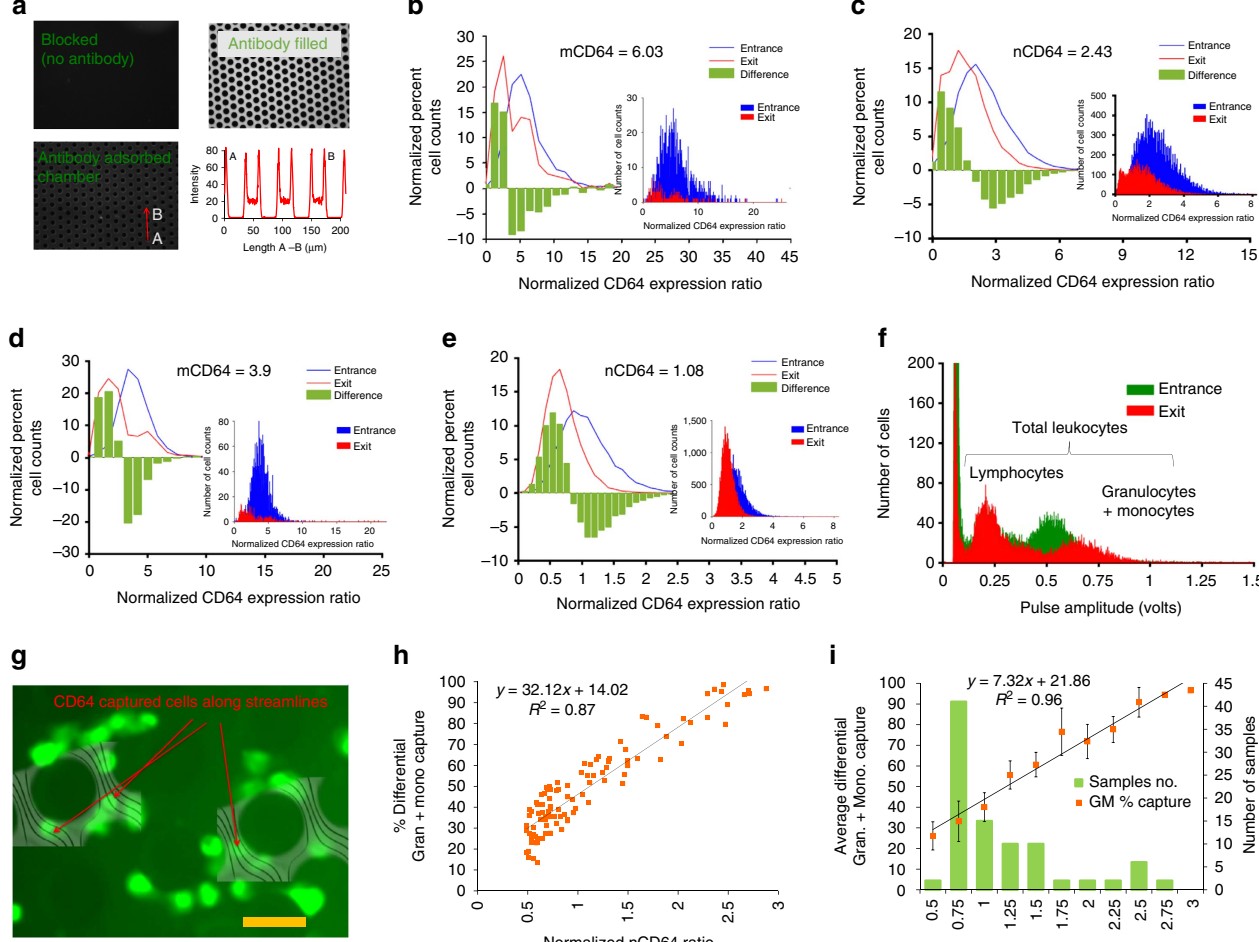

**Figure 3 | Expression-based CD64 cell capture in the capture chamber immobilized with anti-CD64 (clone 10.1) antibody.** (**a**) Antibody adsorption characterization of the capture chamber. BSA-blocked chamber shows no fluorescence; however, PE-conjugated anti-CD64 antibody-filled chamber shows high fluorescence. In antibody-adsorbed chamber, white circular patterns at the periphery of the pillars show the adsorbed antibody after the unadsorbed antibody is flushed away. Intensity plot along A–B, with intensity peaks represent the white regions around pillars' periphery and valleys represent pillars. (**b**) CD64 expression-based capture of monocytes using Sample A (mCD64 = 6.03), with inset CD64 histograms of monocytes before capture (Blue) and after capture (Red) shows almost complete capture. The red and blue curves show the exit and the entrance normalized percent monocyte count versus mCD64 expression,respectively, with green bars representing difference of entrance minus exit cell counts. (**c**) CD64 expression-based capture of neutrophils using Sample A (nCD64 = 2.43). (**d**) CD64 expression-based capture of monocytes using Sample B (mCD64 = 3.9). (**e**) CD64 expression-based capture of neutrophils using Sample B (nCD64 = 1.08). (**f**) The amplitude histogram of the cell pulses from the entrance counter (green) and exit counter (red). (**g**) False-coloured fluorescent image showing the captured CD64 cells around the pillars. Cells are captured specifically with antigen–antibody interaction as they lie alongside the streamlines (scale bar: 40 μm). (**h**) The plot shows a linear correlation between percent granulocytes + monocytes capture versus nCD64 expression ratio with (coefficient of determination: $R^2 = 0.87$, $P < 0.0001$). (**i**) The percent granulocytes + monocytes capture averaged from all samples in a bin versus 0.25 nCD64 bins. The plot shows a linear correlation between them with (coefficient of determination: $R^2 = 0.96$, $P < 0.0001$) with error bars representing s.d in cell capture. The green bars of secondary $y$ axis represent the number of samples in each bin.

nCD64 = 2.43), the non-specific capture of monocytes is minimal as shown with inset CD64 histograms of monocytes entrance, that is, before capture (Blue) and exit, that is, after capture (Red). The red and blue curves show the exit and the entrance normalized percent monocyte count versus mCD64 expression, with green bars representing difference of entrance minus exit cell counts. The slight difference in blue and red curves represents the minimal capture of <2% cells, with highest CD64 expressing cells getting non-specifically attached to the pillars. Presence of more antigens on the cells' surface will result in increased probability of cells getting non-specifically captured. Similarly, Supplementary Fig. 16b shows the non-specific capture of neutrophils of Sample A in the same blocked chamber. The non-specific capture is minimal <5%, mainly of highest expressing CD64 cells. We selected another blood sample 'B' with comparably lower CD64 expression values (mCD64 = 3.9, nCD64 = 1.08) and performed the non-specific cell capture analysis in a blocked chamber. Supplementary Fig. 16c shows the non-specific capture of monocytes of Sample B in a blocked chamber. The non-specific capture is minimal <5%, and of highest expressing CD64 cells. Similarly, Supplementary Fig. 16d shows the non-specific capture of neutrophils of Sample B in the same blocked chamber. Non-specific capture of combined granulocytes + monocytes population using samples A and B is also shown in Supplementary Fig. 17a,c respectively. For another sample C, with low nCD64 value of 0.4, the non-specific capture is minimal as shown in Supplementary Fig. 18a.

We investigated the non-specific cell loss in a blocked chamber using our electrical differential biochip too. Supplementary Fig. 19a shows the representative amplitude histogram of the cell pulses from the entrance counter (green) and exit counter (red). The recovery of the total leukocytes is 98% representing only 2% loss of cells in the capture chamber. We also compared both chamber designs using our biochip as well. We ran blood samples on the chip using the blocked chamber with $n = 35$ blood samples at different flow rates and compared exit versus entrance total leukocyte counts (Supplementary Fig. 19b). Results show that the cell recovery is 98% with 2% loss in the chamber with $R^2 = 0.99$. When the exit versus entrance total granulocyte + monocyte counts were compared, we found the recovery of 93% with 7% loss of cells in chamber with $R^2 = 0.99$ (Supplementary Fig. 19c). However, the exit versus entrance total lymphocyte counts show the recovery of 127% with $R^2 = 0.90$ in Supplementary Fig. 19d. The increase in the exit lymphocyte counts can be associated with the permeabilization of granulocytes and monocyte with extra exposure time to the lysing and quenching buffers.

**CD64 expression-based cell capture in an antibody chamber.** Before investigating the cell capture, we characterized the chamber to ensure the antibody adsorption in the chamber around the pillars. Figure 3a shows the antibody adsorption characterization of the capture chamber. PE-conjugated anti-CD64 antibody is adsorbed in the chamber (antibody adsorption protocol is given in Methods section) and is imaged under fluorescence microscope to detect presence of adsorbed antibody. As a control, a BSA blocked chamber is also imaged and it shows no fluorescence and PE-conjugated anti-CD64 antibody filled chamber shows high fluorescence. After the unadsorbed antibody is flushed away using $1 \times$ PBS, the chamber is imaged again and shows white circular patterns at the periphery of the pillars showing the adsorbed antibody. Figure 3a also shows the fluorescence intensity along the A–B line for three pillars as shown. The peaks intensity represents the white regions around pillars' periphery and valleys represent the pillars.

Figure 3b,c shows the CD64 expression-based capture of cells using a representative high CD64 expressing blood sample 'A' with mCD64 = 6.03, nCD64 = 2.43. In Fig. 3b, the capture of monocytes is maximum as shown by the inset CD64 histograms of monocytes' entrance, that is, before capture (Blue) and exit, that is, after capture (Red). The positive green bars show that the low CD64 expressing cells are the majority of the exit CD64 monocyte population, while most of the high CD64 monocytes got captured. Presence of more antigens of the cells' surface will result in increased probability of cells getting captured. Similarly, Fig. 3c shows the CD64 expression-based capture of neutrophils of sample A in the same antibody chamber. For neutrophil population, high expressing CD64 cells got captured while low expressing CD64 did not. We did the same study using the blood sample 'B' with comparably lower CD64 expression values (mCD64 = 3.9, nCD64 = 1.08). Figure 3d shows the CD64 expression-based capture of monocytes of sample B in the capture chamber, with inset CD64 histograms representing the maximum capture of cells. Similarly, CD64 expression-based capture of neutrophils of sample B is shown in Fig. 3e, with inset CD64 histograms showing less CD64 cell capture owing to the lower nCD64 value. However, the cells which were captured have high CD64 expressions. Supplementary Fig. 17b,d shows the selective capture of CD64 granulocytes + monocytes as a combined population from samples A and B, respectively. CD64 expression-based capture of granulocytes using sample C with nCD64 value of 0.4 shows the minimal capture in Supplementary Fig. 18b. Furthermore, we have done the CD64 expression-based cell capture study using a flow cytometer on 25 patient blood samples (Supplementary Fig. 20). The detailed experimental protocol included off-chip blood lysing and quenching, then flowing cells through blocked and antibody chamber in parallel. The exit cells from both chambers are labelled with fluorescent antibodies; then samples are incubated in the dark for 20 min with subsequent flow through the flow cytometer. The cells are counted from blocked ($B$) and antibody chambers ($A$). The cell capture is defined as $(A–B) \cdot B^{-1}$, normalized to the blocked chamber (Supplementary Fig. 20a). The average differential cell capture (from three replicates) versus nCD64 ratio ($n = 25$ samples) shows a linear correlation of nCD64 expression with increase in the cell capture (Supplementary Fig. 20b). Supplementary Fig. 20c shows the nCD64 expression value based on cell capture (obtained using threefold cross-validation with 500 random trials) versus normalized nCD64 ratio. In Supplementary Note 1, we have also presented a brief theoretical model outlining the direct relationship of increase in antigen density (expression level) on the cell's surface with increased probability of cell capture[33–35].

**Clinical study of biochip for CD64 expression quantification.** We investigated the CD64 expression-based cell capture in an antibody-adsorbed chamber using our electrical differential biochip too. We ran $n = 102$ blood samples collected from patients admitted to Carle Hospital, Urbana and counted their granulocyte + monocyte counts in entrance and exit counters of the biochip. The representative amplitude histogram of the cell pulses from the entrance counter (green) and exit counter (red) is shown in Fig. 3f. The lymphocytes did not get captured; however, almost 50% of granulocytes + monocytes got captured in the chamber. The chamber is collected after the end of experiment, and captured cells are labelled with anti-CD64 fluorescent antibody. After washing away the unadsorbed antibody, the chamber is imaged under fluorescent microscope. Figure 3g shows the false-coloured fluorescent image showing the captured CD64 cells around the pillars. We ran 102 blood samples on our biochip and compared the percent granulocytes + monocytes

capture versus nCD64 expression ratio (Fig. 3h). The plot shows a linear correlation between them ($R^2 = 0.87$), suggesting an increase in capture as the CD64 expression is increased. Bins of 0.25 nCD64 size are created and the percent captured granulocytes + monocytes is averaged of all the samples in the respective bins and plotted against corresponding bin size (Fig. 3i). The plot shows a high correlation ($R^2 = 0.96$) and linear increase in percent cell capture with increase in CD64 expression on neutrophils. The green bars of secondary $y$ axis represent the number of samples in each bin. In precision study, we run multiple experiments using the same samples; the results are shown in Supplementary Fig. 21.

**Patients' stratification and time-course measurements**. The differential capture of granulocytes + monocytes linearly relates with the nCD64 expression ratio. To obtain the nCD64 value from the biochip, we performed threefold cross-validation with

1,000 times random sample selection on the percent granulocytes + monocytes capture versus control nCD64 expression values (data from Fig. 3h). The comparison plot in between nCD64 value obtained from biochip versus flow cytometry shows the good linear correlation of $R^2 = 0.87$ in Fig. 4a. The Bland–Altman analysis shows a high correlation with $-0.0002$ average difference of nCD64 value in between biochip and flow cytometer (control) and 0.49 value as limits of agreement (Fig. 4b). ROC curves in Fig. 4c show the high predictability accuracy for all the three CD64 bins used for predicting sepsis diagnosis (Fig. 2h). AUC for Bins 1, 2 and 3 are 0.98, 0.85 and 0.9, respectively, showing the high predictability accuracy for all the bins. We also calculated the prediction accuracy of nCD64 value from biochip, and we have found that biochip can predict the nCD64 value with greater than 85% accuracy for the minimum nCD64 bin size of 0.6 (Fig. 4d). This corresponds to ten bins when the entire nCD64 range is divided equally. The box plots show the nCD64 expression value obtained from biochip from 91 blood samples

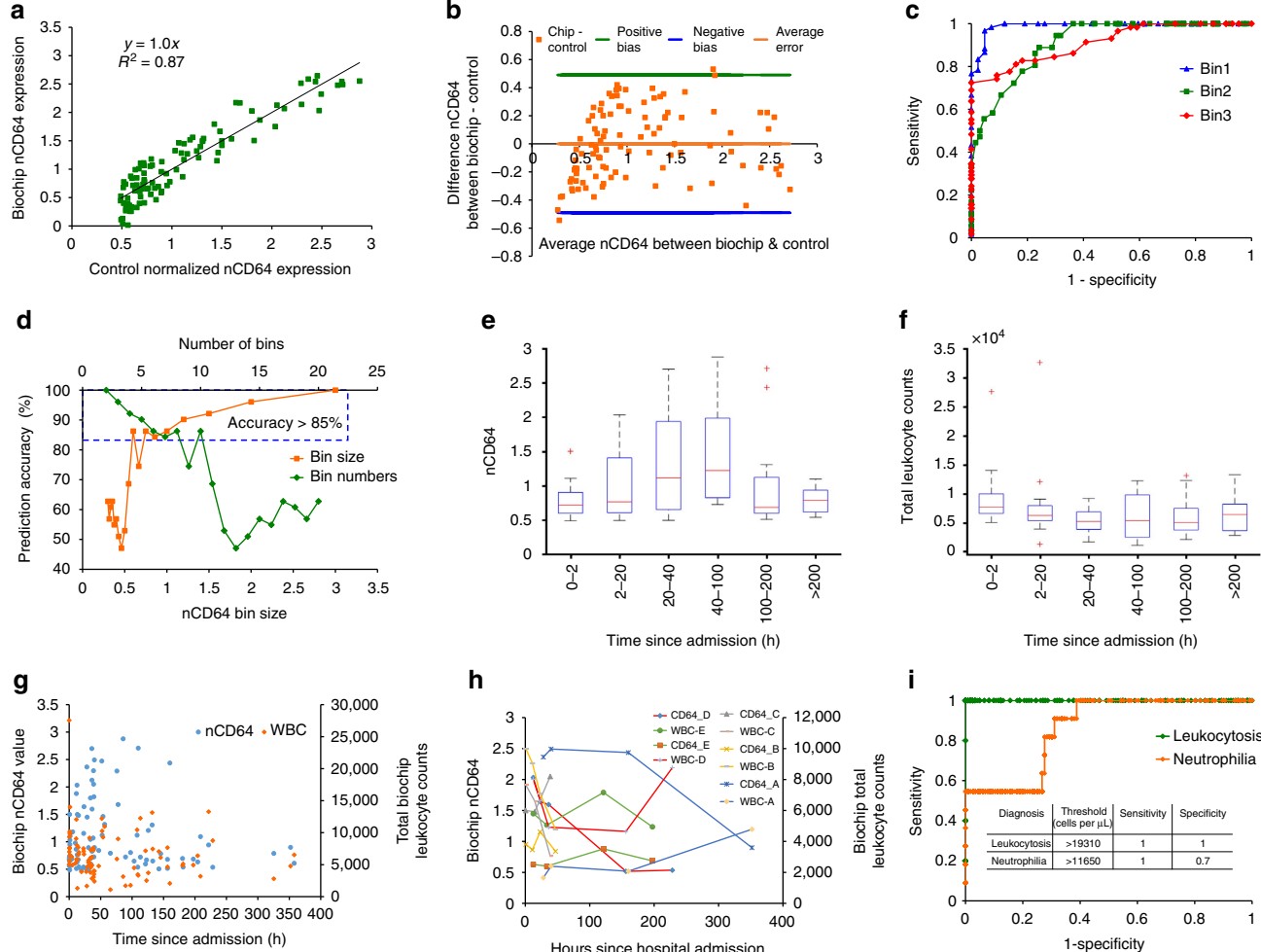

**Figure 4 | Patient stratification and time course measurements.** (**a**) The comparison plot between nCD64 value obtained from biochip versus flow cytometry, showing the good linear correlation of coefficient of determination: $R^2 = 0.87$, $P < 0.0001$. (**b**) The Bland–Altman analysis comparing the nCD64 values from biochip versus flow cytometer (control). It shows $-0.0002$ average difference of nCD64 value in between biochip and flow cytometer (control) and 0.49 value as limits of agreement. (**c**) ROC curves to predict the three bins used for sepsis diagnosis (Fig. 2h). AUC for Bins 1, 2 and 3 are 0.98, 0.85 and 0.9, respectively, showing the high predictability accuracy for all the bins. (**d**) The plot represents the nCD64 prediction accuracy from the biochip. The accuracy is $> 85\%$ for the bin size of 0.6 nCD64 value, which corresponds to ten bins. (**e**) The box plots showing the nCD64 expression value obtained from biochip from 91 blood samples collected from patients at different times of their hospital stay. (**f**) The box plots showing the total leukocyte counts obtained from biochip from 91 blood samples from the same patients as in **e**. (**g**) The plot representing the nCD64 value and total WBC counts obtained from biochip using 91 blood samples collected at different times of patient hospital stay. (**h**) Total leukocyte count and nCD64 value of the patients are tracked over time. (**i**) ROC curves showing high sensitivity and specificity of leukocytosis and neutrophilia diagnosis from the biochip.

collected from patients at different times of their hospital stay. These patients got recovered as also indicated by the increase and then decrease in the nCD64 expression value (Fig. 4e). The box plots showing the total leukocyte counts obtained from biochip are obtained from the same samples. Total leukocyte counts of these patients reached a value of ~5,000 cells per μl in TW6 comparable to a healthy individual (Fig. 4f). The box plots showing the normalized ratio of nCD64 to WBC obtained from biochip are shown in Supplementary Fig. 22. The nCD64 value and total WBC counts obtained from biochip from all 91 blood samples are plotted in Fig. 4g against different sample collection times during the patient stay in the hospital. Total leukocyte count and nCD64 value of the patients are tracked over time. Patients recovered; for example, Patient A has very low WBC count (~1,000 cells per μl) and very high nCD64 value (2.6) at TW1; however, by TW6 their WBC counts became normal (~5,000 cells per μl) and nCD64 value reduced to <1 (Fig. 4h). The biochip is also capable of diagnosing leukocytosis and neutrophilia from the blood samples of patients. ROC curves showing high sensitivity and specificity of leukocytosis and neutrophilia diagnosis from the biochip are shown in Fig. 4i. Cell threshold values and the sensitivity and specificity values are also shown in the inset table of Fig. 4i.

## Discussion

The increase in the CD64 expression is activated by either pro-inflammatory cytokines' interferon gamma (IFN-g) or also granulocyte colony-stimulating factor that are produced in response to the pathogen infection or exposure to endotoxin[36]. The increase in the CD64 expression on neutrophils in septic cases is higher and more specific than other inflammations like autoimmune diseases. Allen et al. showed a twofold increase in the median CD64 expression in patients with inflammatory autoimmune disease as compared to control patients[37]. However, increase in the CD64 expression in the patients with systemic infections was almost seven times higher than that in the control group[37]. Furthermore, neutrophil CD64 expression may also increase after major trauma[38] and sterile insult after major surgery[39]. Neutrophil activation has also been reported to increase in burn and trauma patients and many have investigated their utility to migrate towards infections[40,41]. Similarly, changes in the nCD64 expression have also been reported in patients with postoperative infections who are undergoing vascular[42], cardiac[43] or musculoskeletal surgeries[44]. Some other diseases like inflammatory bowel disease[45] and familial Mediterranean fever[46] have also reported neutrophil activation and have shown changes in the expression level of nCD64.

nCD64 has also been reported to distinguish between bacterial versus viral infections, with higher changes in expression in patients with bacterial infections[47]. Another study has reported to combine nCD64 and expressions of CD35 biomarkers and were able to distinguish between bacterial, viral and other inflammatory diseases[48]. Furthermore, changes in the expression level are mostly age independent. There are numerous studies that have been conducted investigating the utility of nCD64 as biomarker for sepsis detection in neonates and infants. A meta-analysis[8-10] investigating nCD64 utility as a biomarker has shown the sepsis detection cutoff values of nCD64 to be independent of neonates, infants and adults. The increase/decrease in expression can be related to microbe or severity of infection and not to age or gender.

The pathogenic infection that results in pro-inflammatory response and activates CD64 expression on neutrophils also results in activating a few other biomarkers including neutrophil CD11b cells, mHLA-DR on monocytes and certain specific lymphocyte populations (CD4+ and CD25+)[49-52]. Although just combining CD64 with the current stratification technique, that is, SIRS + and suspect or proof of infection, improves the septic predictability as shown by our current work (Fig. 2) and also by numerous studies done previously, we believe that when CD64 is combined with other mentioned biomarkers, we can further improve the accuracy in diagnosing sepsis. Although this work has been focussed on quantifying nCD64 expression on chip, our work can easily be translated to quantify any other cellular antigen expression levels.

There are a very few studies reported that used nCD64 as a prognosis tool for septic patients, because of the inherent difficulty of performing nCD64 quantification assay using a flow cytometer for the length of the patient's stay. Furthermore, the available studies were only able to do nCD64 assay once a day. For example, Dimoula et al. performed serial CD64 measurements daily on septic patients undergoing different antibiotic treatments, and they have shown the correlation of decrease in CD64 expression with the appropriate antibiotic treatment[7]. Our microfluidic biochip provided a unique ability to perform the time course and rapid blood cell counts and nCD64 measurements over the length of patients' stay especially during the different time windows.

Our improvement in AUC from 0.70 to 0.77 is statistically significant as indicated by the p value (<0.001) reported in the Methods section. Much of the previous studies were done in very restricted settings like ICU, where the patients have already confirmed bacterial infections, and so on. However, in our study, we started with a very diversified and heterogeneous patient population having several different infections, and were able to see the statistical improvement in prediction of sepsis. However, a complete random and blinded patient recruitment criteria without any prior sample stratification would be an ideal benchmark for the utilization of our biochip in sepsis prediction.

Developing a PoC system for early stratification of sepsis will have a significant impact on patients and hospitals as delay in the early diagnosis of sepsis can have multiple drawbacks. For example, with the suspicion of infection, physicians usually recommend broad-range antibiotic treatments. Furthermore, inappropriate use of antibiotics, if sepsis and/or infection is not diagnosed, is harmful for the patients as it will interfere with the patient's normal microbiota, and potentially develop antimicrobial resistance to drugs[40]. An early sepsis stratification system could lead both to improved screening techniques and more therapeutic options and potentially drastically reduce lengthy stays in critical care units, priced at >$20,000 per stay on average for septic patients in the US[53]. Furthermore, as the 'Pay for Performance' initiative is implemented in hospitals across the United States, their reimbursement is based on showing steady improvement over time in critical statistics including lengths of stay, mortality rates, rates of hospital-acquired infections and many others. Sepsis has been identified as a Core Measure because better adherence to standardized sepsis guidelines has been shown to drastically improve all of these critical parameters. Therefore, we believe that a more specific and sensitive method to perform accurate stratification of sepsis will not only improve patient outcomes but will be a critical part of the financial survival of hospitals across the United States.

Our CD64 quantification biochip can easily be adapted for other applications. Changes in the expression level of cellular antigens play a critical role in pathogenesis and inflammation, and they can be investigated by clinical studies. The different type and level of antigen expression is helpful for disease diagnosis and monitoring its progression, and can be used for phenotypic cellular measurements. We can easily translate our biochip for other applications including quantifying the CYP1A1 and

CYP1B1 expressions in human endothelial cells to understand their role in pro- or anti-atherogenic endothelial cell functions[54]. In meningiomas, increased expression of Ki67 is a predictor of not only high-grade malignancy but also recurrence in gross-totally removed benign meningiomas[55]. Another example is expression of CD71, the transferrin receptor, in identifying CTCs activated during tumorigenesis. CD71 is associated with cell growth regulation and iron uptake and is expressed on low levels in resting cells, whose expression increased on proliferating cells[56].

CD64 expression is a proinflammatory biomarker at the onset of infection. Therefore, a quick PoC testing (different settings like bed-side, or emergency for trauma and/or injured patients) can have an important clinical impact in evaluating the immune system of the patients at the onset of their infection. Currently, available benchtop flow cytometers with reduced complexity, for example, by companies like CytoBuoy and Cronus Technologies, as well as by larger companies such as Sysmex and Abbott, still require traditional flow cytometry sample preparation to be performed in a microbiology lab before the sample can be introduced to the device. Any blood diagnostic test process flow in the hospitals is prolonged, starting with physician placing a blood test order, phlebotomist coming to the patient room, drawing blood and sending it to flow lab for later processing. And during this time, the diagnostic window of opportunity can potentially be missed. Here, we believe that our PoC system can have a clinical impact that it will bypass the entire current process flow and provide the testing with immediate availability of results within a matter of minutes to the physicians.

Our current CD64 expression quantification experimental assay takes about 30 min to complete. This time can be greatly reduced by increasing flow rate and making some geometry changes in the biochip (Supplementary Table 4). By changing the geometry (height and width) of the capture chamber, we can keep the similar shear stress conditions for antigen–antibody interactions on the pillars surfaces. The detailed description on flow rate characterization and its effects on sensitivity of the counter and throughput and the required blood volume are given in Supplementary Notes 2 and 3.

In conclusion, we have demonstrated a robust design of a biochip for potential stratification of sepsis in patient population at Carle Hospital. We have shown a point-of-care experimental assay to quantify CD64 antigen expression and leukocyte count from whole blood without the need of any manual sample processing. We have validated our PoC biochip by measuring cell counts ($n = 181$) and nCD64 levels ($n = 102$) from 10 μl of patients' blood samples and have shown excellent correlation with flow cytometer. The targeted patient population is very diverse with respect to cell counts (dynamic range: 530–38,570 leukocytes per μl) and antigen expression levels, and our biochip shows excellent measurement accuracy for the entire dynamic range of these parameters. Furthermore, we have shown the biochip's utility for sepsis diagnosis and prognosis by performing these measurements on patients' blood samples collected at different times of their stay at hospital. In future, this biochip can potentially be used at the patient's bedside for continuous monitoring of patient's immune system in response to different therapeutic interventions at different stages of the disease.

## Methods

**Chamber immobilization protocols.** *Blocked chamber.* 1% BSA in $1 \times$ PBS is used as a blocking solution. Capture chamber is filled with the blocking solution at the flow rate of 30 μl min$^{-1}$ and then it is let to be adsorbed for a minimum of 30 min.

*Antibody chamber.* A stock solution of anti-CD64 antibody is made with 0.5 g l$^{-1}$ by mixing 500 μg of lyophilized antibody with $1 \times$ PBS and storing it at 4 °C. For each chamber preparation, antibody solution is prepared by diluting the stock solution with $1 \times$ PBS at a concentration ratio of 0.23, that is, stock solution: PBS (v/v). Chamber is filled with this solution and let it sit for 30 min, and procedure is repeated again; this allowed sufficient time for antibody adsorption. Unadsorbed antibody is then flushed away using $1 \times$ PBS flowed at 10 μl min$^{-1}$. Antibody adsorbed chamber is then filled with blocking solution with a subsequent wait time of 30 min before using the chamber in the experiment.

**Reagents.** *Lysing solution.* The composition of lysing solution is 0.12% Formic acid (v/v) and 0.05% Saponin (w/v) in DI water.

*Quenching solution:* The composition of quenching solution is 21.1785% $10 \times$ PBS (v/v) and 0.57525% sodium carbonate (w/v) in DI water.

*Antibodies.* Unconjugated anti-human CD64 antibody (Clone: 10.1, Cat: MAB1257-100, R&D Systems) and PE anti-human CD64 antibody (Clone: 10.1, Cat: 305008, BioLegend) are used.

**Biochip fabrication protocols.** Fluidics layers fabrication: The master mould (negative) of the fluidics regions of the biochip including lying module, counter module and capture chamber was created by a standard SU8-photolithography process using SU8-50 negative photoresist on Si wafer. The heights of the counters module, lysing/quenching region and capture chamber were 15, 90 and 60 μm, respectively. The surface of the mould was silanized by 3-mercaptopropyl-trimethoxysilane. The actual biochip was made out of polydimethylsiloxane, in which elastomer is mixed with curing agent at the ratio of 10:1 with subsequent pouring on the master mould. After removing all bubbles using desiccator, it is cured at 90 °C for 30–60 min. Electrodes' fabrication: The Pyrex wafer is patterned using LOR3A and S-1805 photoresists to make a negative pattern of the electrodes. CD-26 developer is used to develop the wafer. Titanium (25 nm) (adhesion layer) and 75 nm of platinum (metal layer) are evaporated onto the wafer. Microchem PG remover is used to remove unwanted metal from the wafer for 20 min at 70 °C treatment[30].

We tested each counter for quality control before using in the experiment. Test of the counter includes testing for any shorts after applying the conductive epoxy and also measuring resistance from the electrode to the PCB lines. We have used <5 Ohms resistance in between the electrode and PCB connecting lines as a control parameter.

**Experimental setup and instrumentation.** The input buffers (lysing solution, quenching solution and $1 \times$ PBS as push buffer for blood) are infused into the biochip using an Eksigent Nanoflow LC pump. Ten microlitres of blood is metered in a 0.012 inch inner diameter PTFE tubing and placed into the blood input of the biochip. Blood, lysing and quenching buffers are infused at 0.518, 6.218 and 3.264 μl min$^{-1}$, respectively. Blood: lysing ratio is 1:12 and blood: quenching ratio is 1:6.3. Total on-chip lysis time is 6.1 s and lysed blood is quenched for 38.24 s. Zurich Instruments HF2LI lock-in amplifier is used to provide the signal input to the electrodes (303 kHz, 5Vp-p) to both counters. The output voltage signals from the electrical counters are fed to the differential preamplifier HF2CA, which in turn are fed to the HF2LI for further noise removal. The differential signal was obtained using a built-in module of the lock-in amplifier HF2LI (Supplementary Fig. 4). DAQ card (PCI-6351, National Instruments) is used to acquire the output signal from lock-in amplifier and data is stored into the computer at the sampling rate of 250 kHz. Data analysis including pulse detection and cell counting was done using a customized program written in Matlab. The electrical counter needs to detect the particles of different sizes flowing through it. The pulse amplitude histogram of the 5.5 and 7 μm beads flowing through the electrical counter is shown in Supplementary Fig. 23, showing that the counter is sensitive enough to differentiate between these two bead populations.

**Electrical cell-counting analysis.** The acquired data from the experiment were analysed in a customized program written in Matlab. As the cell passes through the aperture, it generates the voltage pulses, which were samples at 250 kHz. The cell-counting data are shown in Supplementary Fig. 5a before performing any digital filtering. The low-frequency noise including the baseline drifts were removed using the high-pass filter with a 20 Hz cutoff frequency. Power line interference of 60 Hz and its first harmonic (120 Hz) are removed by using two band stop filters with cutoff frequencies of (58, 62) Hz and (118, 122) Hz, respectively. The input frequency noise of 303 kHz was removed by using a low-pass filter with 303 kHz as a cutoff frequency. The data after performing the digital filtering is shown in Supplementary Fig. 5b. The maximum amplitudes of all the voltage pulses are obtained by comparing with a threshold level of 0.5 V, which is approximately 10 × standard deviation of baseline noise. The thresholding in between lymphocytes and granulocytes + monocytes is done considering minima in between two populations.

Supplementary Fig. 24 compares the electrical signals obtained as cell, and possibly micro debris particle (generated due to the lysis of red blood cells) passes through the electrical counter. The figure shows the representative electrical voltage signal of a cell passing through the counter and the dotted rectangular region, which is zoomed-in to show the signal possibly related to the microdebris particles flowing through the counter channel. Furthermore, pulse width is calculated by

measuring the number of samples in the dotted region and then multiplying it by sampling time as shown in Supplementary Fig. 25.

**Blood sample acquisition.** Clinical studies of the biochip were done at Carle Hospital, Urbana, IL. Blood samples were obtained from the anonymous patients who are SIRS-positive and/or their blood culture is ordered by a physician with suspicion of an infection. The left-over blood samples from patients after laboratory tests were de-identified by the clinical lab staff and sent to us. Samples were collected until their discharge from the hospital (recovery or death) at different possible time intervals through an authorized University of Illinois Urbana-Champaign and Carle Foundation Hospital Institutional Review Board (IRB) applications (UIUC IRB number 15500 and Carle IRB number 15014). EMR data not considered protected health information (PHI) for patients who were discharged or deceased were retrieved by an authorized personnel and made available to the PI with approved de-identification protocol. According to the University of Illinois at Urbana-Champaign (UIUC) and Carle Foundation Hospital's IRB guidelines, patients were not informed of the diagnostic results from our device or from the flow cytometry controls. Blood samples were collected in vacutainers coated with EDTA and are kept on a rotisserie at room temperature. Experiments are conducted within 12 h of blood samples' availability.

**CD64 control expression protocol.** The stepwise blood-processing protocol is shown in Supplementary Fig. 9a. Eight microlitres of whole blood is lysed with 96 μl of lysing reagent for 6s. Further 50.4 μl of quenching reagent is mixed with the lysed blood. Eight microlitres of cocktail of conjugated antibodies (FITC anti-CD64 and PE anti-CD163 from Trillium Diagnostics kit) is added to the solution. The sample is incubated for 20 min. Finally, 5 μl of FITC-conjugated control beads was added to the solution, serving as an internal control. Supplementary Fig. 9b (left) shows the forward and backscatter plot of a sample obtained after running the sample on a flow cytometer. There is a clear differentiation between total white blood cells, debris and the control beads. The WBC and the bead populations were gated and represented by gates R1 and R5, respectively. Supplementary Fig. 9b (middle) shows the scatter plot of the 575 (PE) and backscatter channels. The expression of CD163 on the PE channel was used to differentiate between monocytes, lymphocytes and neutrophils by their respective gates. The histogram of the cell populations is obtained on the FITC (530) channel representing the CD64 fluorescence intensity. Mean fluorescent intensity (MFI) is obtained for monocytes, lymphocytes and neutrophils. CD64 FITC-conjugated beads in the sample serve as an internal control and the MFI of the beads is used to normalize MFI of the leukocyte subpopulations. Supplementary Fig. 9c (right) shows the CD64 FITC histogram of monocytes, neutrophils and lymphocytes on the 530 channel of the flow cytometer.

Guava EasyCyte Plus flow cytometer was used for all the CD64 control measurements. Guava was calibrated daily before performing any control assay using a standardized manufacturer protocol (ensuring that coefficient of variation of control beads counts and their fluorescence measurements is <5%). A 96-well plate is used to perform CD64 measurements on multiple samples at once. The gating on the scatter plots for differentiating cell populations is done by FCS Express software or using our custom-written Matlab program.

**COMSOL simulations for chamber design.** Capture chambers were simulated using COMSOL v5.2. We have developed a multiphysics model using 'Laminar flow' and 'Particle Tracing for Fluid Flow' modules in COMSOL. The chamber structure was meshed using 'Fine' category as element size. No slip boundary condition is selected for the particle interaction with chamber walls. We have simulated particle sizes of 8 and 13 μm corresponding to the average diameters of lymphocytes and granulocytes + monocytes population. Different chamber designs were simulated with variable staggering ratio of the pillars' rows comparable to their subsequent previous rows. The flow streamlines of the respective particles were also plotted, which showed variability in the zero-velocity or dead-zone regions around the pillars attributing towards non-specific entrapment of cells in the capture chamber. Using 'release time' and 'stop time' output parameters, we were also able to calculate the traversal time for the particle to flow through the simulated chamber for all the individually released particles. This allowed us to find how many particles got stuck in the dead zones of the chamber and optimize the chamber design.

**Design of a capture chamber.** We designed the capture chamber on chip to ensure that, first, the cells experience maximum time interacting with the pillars, where antibodies are adsorbed. Second, to minimize cell capture in the zero-velocity or stagnation regions around the pillars. Third, different sized cells should experience the same amount of interaction time in the chamber. Therefore, we simulated different designs of the capture chambers in COMSOL with 8 and 13 μm diameter particles flowing between the pillars. To enhance the cell–pillar interaction in a uniform way, the subsequent pillars' rows were staggered at different ratios as compared to the previous rows. Row shifting was calculated by staggering ratio × (pillar–pillar) spacing, as shown in Supplementary Fig. 26. In our previous study of CD4 T-cell capture, we have used the capture chamber with the staggering ratio of 0.5. However, here the chamber design was improved significantly, as per

the above three criteria which are critical for expression-based cell capture and accurate cell counting. We simulated the chambers with different staggering ratios of 0, 0.1, 0.2, 0.3, 0.33, 0.35, 0.4 and 0.5 in COMSOL. Supplementary Fig. 27 shows the COMSOL images of simulated capture chambers with different pillars' staggering ratios with flow streamlines around the pillars. For a chamber with staggering ratio of 0, if a particle is flown in the middle stream, it may never interact with a pillar. However, for a chamber with a staggering ratio of 0.33 or 0.5, streamlines shift their positions in between pillars, thereby making sure a particle will interact with the pillar more uniformly irrespective of the size. We fabricated the chambers with pillars' staggering ratios of 0.142, 0.2, 0.25 and 0.33 and flow beads through it for the experimental visualization of the beads–pillars interaction. Supplementary movies 1–4 show the recorded videos of the beads interacting with the pillars after every 7, 5, 4 and 3 subsequent rows with staggering ratios of 0.142, 0.2, 0.25 and 0.33, respectively. The simulated peak transit time and transit time difference between 8 and 13 μm diameter particles versus pillar staggering ratio is shown in Supplementary Fig. 28a. It shows that for 0.33 staggering ratio of pillars, the 13 μm particles experience the highest transit time, indicating more interactions with the pillars. Furthermore, we have also compared the number of stagnated particles at zero-velocity regions in different chamber designs. This is shown in Supplementary Fig. 28b for both 8 and 13 μm particles. Chamber with the ratio of 0.33 shows highest transit time for the 13 μm particle (corresponding to granulocytes + monocytes); however, the chamber with 0.35 ratio has the minimum difference between transit time of 13 and 8 μm particles. Between 0.33 and 0.35, we selected to use 0.33 chamber design as it has 3.5% stagnated particles as compared to 6% in 0.35 design. We also compared the two chamber designs, with Designs 1 and 2 having pillars' staggering ratios of 0.5 and 0.33, respectively. Supplementary Fig. 29a,b shows that the particles (8 μm diameter) flowing through the Design 2 chamber have a transit time profile, indicating more uniform interactions with the pillars as compared to Design 1 chamber. The Design 2 chamber showed higher transit time for 13 μm particles as compared to Design 1 chamber, as shown in Supplementary Fig. 29b. Supplementary Fig. 30a shows the images of simulated capture chambers with pillars staggering ratio = 0.33 with flow streamlines around the pillars. The dotted red circles show the potential stagnation (zero-velocity) regions where the particles are trapped (Left: 13 μm particles, Right 8 μm particles). Supplementary Fig. 30b shows the image of a blocked capture chamber and cells stuck at zero-velocity regions around the pillars. Simulated streamlines have been overlaid on nearby regions for easier visualization of the stagnated region around the pillars.

Furthermore, Supplementary Fig. 31 shows the box plots representing accuracy as percent absolute difference in total leukocyte counts from entrance and exit counters while comparing Designs 1 and 2 chambers using $n = 8$ blood samples, respectively. The average percent difference for Designs 1 and 2 chambers is 12.92 and 3.94%, respectively, suggesting that pillars staggering of 0.33 worked best as compared to those of 0.5, thus also confirming our simulated results.

**Study design.** *Power analysis.* We used power analysis to determine the sample size to get the desired statistics. To get the correlation coefficient, $\rho = 0.9$ in between the nCD64 values from biochip versus control, an alpha level (two-tailed) of 0.01 and power at 0.9. For $\rho = 0.9$, $Z_\alpha = 2.58$. However, $Z_b = 1.28$ for the desired 0.9 power level. The following equation is used to calculate the required sample size, which gives $N = 13$.

$$N = \frac{(Z_\alpha + Z_b)^2}{\frac{1}{4}\left[\log_e\left(\frac{1+\rho}{1-\rho}\right)\right]^2} + 3.$$

*Blinded samples and experiments.* According to the IRB protocol, the samples were taken anonymously, such that the authors had no knowledge of the donor's identity, age, gender or ethnicity. De-identified EMR information of the patients was only made available to the authors after the patients' discharge from the hospital through an institutional approved process. Furthermore, assessment of the outcomes (cell counts and nCD64 values) was blinded as the flow cytometry control measurement results were available after the biochip experiments were completed. Data analytic model (SVM), that is employed to predict sepsis in patients used a tenfold cross-validated SVM algorithm. Model is trained using 9/10ths of the data (known cases) and then tested on the remaining 1/10ths (blinded cases). The process is then repeated 1,000 times, each time training and testing on a random permutation of instances.

**Statistical analysis.** Bland–Altman analysis: It was used to compare the agreement between the two nCD64 measurement methods (biochip versus flow cytometer). It a plot of the difference of nCD64 value between two methods versus the average nCD64 of both methods. It provides a bias value, which is the average error (nCD64 difference) between both methods. The positive and negative limits of agreements were calculated as bias value ± 1.96 × SD of the difference. Two-tailed P values were also calculated. For regression analysis, $P < 0.0001$ rejected the null hypothesis, that is, there exists no correlation between biochip and control cell counts. Statistical analyses (regression and Bland–Altman) were performed by data analysis add-in for Microsoft Excel. Cross-validation: It is commonly known as rotation estimation and was used to train the model to obtain the biochip nCD64 value from percent captured granulocytes + monocytes. We used

threefold cross-validation meaning partitioning data into three equal subsets and performed analysis (linear correlation) on two sets combined (training set) and validating analysis on the third (testing set). To reduce variability, we have done 1,000 rounds of cross-validation by randomly selecting samples in the three subsets and averaged the results over all rounds. Box plots: We have used Matlab to plot box plots in this study. The red-coloured cross-bars represent the outliers, which are selected if the data points are greater than $q3 + 1.5 \times (q3 - q1)$ or less than $q1 - 1.5 \times (q3 - q1)$, where $q1$ and $q3$ are the 25th and 75th percentiles of the sample data, respectively. Prediction model: We used SVM as a statistical method to develop the model for sepsis prediction. We ranked features/groups in the model according to their weight vector coefficients. ROC curves are generated to determine whether mean ranks of populations for each ROC curve (each group) significantly differ, which was reported by the $p$ value. Furthermore, the variation in the features sets, for example, age, gender, chronic conditions and infections, can be seen as percent populations in septic and non-septic groups, as shown in Supplementary Table 2. The coefficient of variation (CV) for total leukocyte counts for septic and non-septic groups is 62.88 and 68.81%, respectively. Furthermore, the CV for nCD64 value for septic and non-septic groups is 46.07 and 56.60%, respectively.

**Artificial neural networks (ANN) analysis.** We have used a neural network toolbox in Matlab and used pattern recognition to develop a model for predicting the patient recovery outcome at different time windows for SIRS-positive patients. We have used a two-layer feed-forward network with sigmoid-hidden and softmax output neurons. We simulated using ten neurons in hidden and one neuron in output layers. Input to the ANN model was total WBC count and nCD64 value at their respective time windows, and targets of the model are the patients' outcomes (recovered or died). The data were divided into 50% training, 25% validation and 25% testing samples. The network was trained using scaled conjugate gradient backpropagation method. Network was trained when the generalization stopped improving also indicated by the increase in cross-entropy error. This step was performed by the built-in Matlab program. Hinton diagrams and the table representing the weight and bias values are provided in Supplementary Fig. 15.

**Data analytic model for sepsis diagnosis.** To predict the onset of sepsis, we categorized the data into three feature sets: (1) Quick SIRS, which included temperature, pulse, respirations and systolic blood pressure, and (2) Quick SIRS criteria, lactic acid and biochip parameters including nCD64, total leukocytes counts and its differentials (granulocytes + monocytes and lymphocytes). An attending physician labelled our patients as septic or non-septic. To predict the onset of sepsis, we used a tenfold cross-validated SVM over 1,000 iterations for each of the three feature sets. We train our model using 9/10ths of the data (known cases) and then test the model on the remaining 1/10ths (blinded cases). We then repeat this process 1,000 times, each time training and testing on a random permutation of instances. We also performed a one-sided paired Wilcoxon test between the ROC curves generated to determine whether the difference between the AUC of the ROC curves obtained was statistically significant and whether mean ranks of populations (each ROC curve) for each feature set differ. The $p$-values for the comparison of these two data sets were less than 0.0001.

**Code availability.** The authors declare that algorithms supporting the findings of this study are available within the manuscript and its Supplementary Information files. Furthermore, raw code used in this study is available from the corresponding author upon reasonable request.

**Data availability.** The authors declare that data supporting the findings of this study are available within the manuscript and its Supplementary Information files. Furthermore, raw data of this study are available from the corresponding author upon reasonable request.

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

## Acknowledgements

The authors thank E. Iniguez, V. Reddi, J. Eardley and R. Grubbs of Carle Foundation Hospital for their help in conducting this study. The authors also thank N. Hung, Z. Haidry, C. Hwu, V.F. Bartumeus, G. Tufte, M. Saadah and M. Chheda for their help in devices' fabrication. The authors acknowledge the support of Center for Integration of Medicine and Innovative Technology Innovation (CIMIT)'s Point-of-Care Technology Center in Primary Care (POCTRN) Grant and funding from University of Illinois at Urbana-Champaign.

## Author contributions

U.H., B.R. and R.B. designed the study. U.H. wrote and R.B., B.R., I.T. and T.J. edited the manuscript. U.H., A.T., Z.P., N.M., R.H., A.H., E.F. and J.B. performed the experiments. U.H., I.T., B.R. and A.T. performed the data and statistical analysis. T.G. helped in optimization of capture chamber geometry, electrical counter and its fabrication protocol. T.J. helped in the flow cytometry measurements. M.P., M.R. and S.L. fabricated the devices. J.K., K.W. and B.D. performed clinical patient adjudication.

## Additional information

**Competing interests:** B.R., R.B. and U.H. have financial interests in Prenosis, Inc. All other authors declare no competing financial interests.

