## [Peer review file · Nature Communications]

Reviewers' comments:

Reviewer #1 (Remarks to the Author):

This paper developed a biochip that captures CD64 neutrophils and monocytes on the chip while also counting the other cells that are not captured to provide the physician with point of care (POC) information to predict sepsis. The technology is based on the authors' earlier publications with a few improvements but most significantly the intended application in sepsis with patient samples. The paper is very intriguing clinically and equally important it is a major engineering advance in the manner it is used and quantified. However the paper suffers from several major issues that need to be addressed before publication.

The paper never uses blinded samples and predict sepsis. The samples are all for known cases. There is always unintentional bias in such studies and as such it is hard to rely on an unblinded study to make very strong clinical claims as in this paper. This does not take away from the important contribution of the paper but the authors are urged not to overstate the findings. In addition, the controls are not rigorous. There are many other clinical conditions that might increase neutrophil activation (only a few listed in Discussion but those are usually not the most relevant ones). Nevertheless the study is robust enough but the authors should mention these shortcomings explicitly instead of dismissing in Discussion as they do currently. In short, the authors are highly over reaching their data with their clinical claims in the absence of blinded samples and better control samples.

The paper is extremely hard to read and follow as it is highly focused on engineering optimization. Most of the figures can be easily moved to the Supplementary material and focus the paper on its clinical importance. The novelty of the paper is its intended use for sepsis and not the incremental engineering performed over the past papers from the same group in other applications. This will require serious rewriting of the paper but in the long run will make the paper more accessible to those who are more interested in the use of the technology and not the details of the engineering.

On a minor note, the rapid assessment is clearly a major issue in sepsis but using fingerpick of blood is not as the authors state in several occasions. It is relatively easy to draw blood from septic patients in hospitals.

Reviewer #2 (Remarks to the Author):

Hassan et al reported a point-of-care (PoC) device for potential sepsis management using neutrophils (nCD64) as a biomarker. The system was built upon some of the prior work from the same group but here for a different application. The system is interesting and the investigation included theoretical simulation, prototype development, and clinical study. However, the paper has several issues that need to be addressed/considered.

- 1) Patient characteristics including age, gender, type of sepsis/infection, etc should be presented and discuss on whether/how these characteristics might affect the nCD64 being a biomarker for sepsis should be included.
- 2) In Fig 2c, the authors reported the system improved AUC from .70 to .77; Please comment on how significant clinically this might be. Overall, while the general premise of this PoC system for more effective sepsis stratification is recognized, it is less clear how exactly this will improve clinical outcome. Does the new system improve any of these: accuracy, sensitivity, and specificity? And can it translate to improved patient survival, saved cost, etc. A discuss would be useful here.
- 3) Along the same line, especially, since different iterations of flow cytometry has been developed in the past; if only a minute of blood sample is required for each test, cannot we just dilute the

blood with buffer and run through a portable flow cytometry without any "manual processing"? If flow cytometry profiling nCD64 has not made an impact in sepsis stratification, why would we expect this system in the paper to do so? Apart from being "complicated" for flow cytometry (which is debatable), are there any other hurdles that might prevent the presented system from making an immediate clinical impact?

4) "The 72 hour survival rate decreases by 7.7% every hour that appropriate antimicrobial medication is delayed," is misleading, as this only applies to the first few hours at onset of the condition.

5) Is a drop of blood enough for accurate blood biomarker in this case? There has been recent evidence showing small volume blood (10 ul used in the study) could create more variation in blood marker detection. That is why current clinical testing typically uses a larger volume of blood.

6) "The CD64+ cells are captured based on the CD64 expression level on their surfaces."; In order to support this statement, which is key for the study, the authors should provide more extensive theoretical and actual data to support this for a broader range of CD64 expression levels (not just high and low).

Minor comments:

7) The list of references is not comprehensive enough, e.g. the following key reference should be cited: J. Cid et al. 2010. Neutrophil CD64 expression as marker of bacterial infection: A systematic review and meta-analysis. *J. Infection*. 60: 313–319.

8) In Figure 1, lysis buffer was used to break the red blood cells. Is the cell lysis mixture still in the channel? If the debris with comparable size passed through the electrodes and creates impedance changes, how could the system filter out those noises?

9). In Figure 1b, how the pulse width is measured?

10). In Figure 4, the flow rate was very briefly investigated. Capability to handle a broad range of flow rate is important to evaluate the sensitivity of the impedance sensing module and the throughput. A more thorough study on flow rate is highly recommended.

11). Supplementary material, Figure 1, it seems the connection pads are connected to traces on PCB using soldering paste. The traces are long and the detection wires seem not protected by any shield. Would the sensitivity of Impedance changes by micro level particles be compromised by this system setup?

12). Supplementary material, Figure 5, why the x-axis is not time?

13) Fig 2 d and e, and Fig S8 and S9, what are those red color cross bars?

14) There are several typos and grammar issues in the text. The authors should carefully proof-read the manuscript.

15) I am also not sure if the clinical studies performed in the study can be considered as "clinical trials".

Authors Response to the Reviewers Comments:

Reviewer #1:

This paper developed a biochip that captures CD64 neutrophils and monocytes on the chip while also counting the other cells that are not captured to provide the physician with point of care (POC) information to predict sepsis. The technology is based on the authors' earlier publications with a few improvements but most significantly the intended application in sepsis with patient samples. The paper is very intriguing clinically and equally important it is a major engineering advance in the manner it is used and quantified. However the paper suffers from several major issues that need to be addressed before publication.

Our Response:

We appreciate the reviewer's comments that our paper is a major engineering advance and clinically intriguing. As you will see from this response, we have further enhanced the paper based on the reviewer's critical comments and added additional characterizations, results and discussion that provide further clarity and depth to our paper. The comments of the reviewer have made the paper better.

The paper never uses blinded samples and predict sepsis. The samples are all for known cases. There is always unintentional bias in such studies and as such it is hard to rely on an unblinded study to make very strong clinical claims as in this paper. This does not take away from the important contribution of the paper but the authors are urged not to overstate the findings.

Our Response:

We appreciate the reviewer's critical comments. We would like to clarify further:

First, in our statistical models we used cross validation technique. It allows dividing sample population into known and blinded cases with training the model on known samples and trying to predict the blinded ones. Specifically, in the scheme of 10 fold cross validation that we employ (in Figure 2c), we train our model using 9/10ths of the data (known cases) and then test the model on the remaining 1/10ths (blinded cases). We then repeat this process 1000 times, each time training and testing on a different permutation of instances. We have revised the text in Methods Section (Data Analytic Model for Sepsis Diagnosis in manuscript page 25 line 26) to make it more clear for the readers.

Second, we had a much generalized inclusion criteria in our study (SIRS positive and/ or ordering of a blood culture for a patient) which further indicates the robustness of our model. Our patient population had a variety of confounding factors such as the presence of acute and chronic comorbidities, including systemic inflammatory conditions that may influence the level of nCD64 expression and other inflammatory mediators. We have added a Supplementary Table 2 providing more information on patient characteristics (including age, gender, type of infection etc.).

However, we also completely agree with the reviewer that a complete blinded patient recruitment criteria without any prior sample stratification would be ideal benchmark for the utilization of our biochip in sepsis prediction. We plan to do this in our next study. Considering this reviewer's comment, we have revised the claims in the manuscript and tried harder to not overstate the findings. For example, please see the corresponding changes in the manuscript:

Page 24 line 7, page 17 line 8 page 25 line 26.

In addition, the controls are not rigorous. There are many other clinical conditions that might increase neutrophil activation (only a few listed in Discussion but those are usually not the most relevant ones). Nevertheless the study is robust enough but the authors should mention these shortcomings explicitly instead of dismissing in Discussion as they do currently. In short, the authors are highly over reaching their data with their clinical claims in the absence of blinded samples and better control samples.

Our Response:

We agree with the reviewer that the neutrophil activation may increase in certain diseases e.g. autoimmune diseases, however the increase in the activation in particular CD64 is much less as compared to sepsis. Now we have added additional examples of neutrophil activation and CD64 expression changes. Furthermore, neutrophil CD64 expression may also increase after major trauma [Ref. 38] and sterile insult after major surgery [Ref. 39]. Neutrophil activation has also been reported to increase in burn and trauma patients and many have investigated their utility to migrate towards infections [Refs. 40, 41]. Similarly, changes in the nCD64 expression have also been reported in patients with postoperative infections who are undergoing vascular [Ref. 42], cardiac [Ref. 43] or musculoskeletal surgeries [Ref. 44]. Some other diseases like inflammatory bowel disease [Ref. 45] and familial Mediterranean fever [Ref. 46] have also reported neutrophil activation and have shown changes in the expression level of nCD64.

We have added the above discussion and more literature review in the manuscript at page 16 line 2.

The paper is extremely hard to read and follow as it is highly focused on engineering optimization. Most of the figures can be easily moved to the Supplementary material and focus the paper on its clinical importance. The novelty of the paper is its intended use for sepsis and not the incremental engineering performed over the past papers from the same group in other applications. This will require serious rewriting of the paper but in the long run will make the paper more accessible to those who are more interested in the use of the technology and not the details of the engineering.

Our Response:

We thank the reviewers' suggestion. As you will see, we have extensively revised the manuscript and shifted some of the technical information (figures and text) related to technology optimization to Supplementary Information, while refocusing on the clinical results. Here are some of the changes we made:

- 1. Previous, figures 1b, 1c (bottom), 1d are shifted to Supplementary Information.***
- 2. Previous, Figure 3 (a-i) all panels are shifted to Supplementary Information. Furthermore, the associated text on 'Design of a capture chamber and its optimization' is moved to the Methods section.***
- 3. Previous, Figure 4(b-f) associated with the characterization and optimization of biochip with a blocked chamber are moved to Supplementary Information.***
- 4. Now, the discussion section is highly focused on the clinical impact, limitations of the current study, and potential usage of this technology for other clinical applications.***

We hope now the reviewer will agree that the revised version is more focused on the clinical usage of the technology and its underlying results rather than engineering optimization of the technology.

On a minor note, the rapid assessment is clearly a major issue in sepsis but using fingerpick of blood is not as the authors state in several occasions. It is relatively easy to draw blood from septic patients in hospitals.

Our Response:

Yes, we agree with the reviewer that the major issue is the rapid sepsis stratification and the method of blood draw is not a limiting factor especially for the patients admitted to hospitals in particular ICUs. Most of the patients might have the line port already installed in their arms/veins by the nurses for necessary blood draws.

We have revised the text accordingly and also added some discussion in Supplementary Information at page 3 line 17.

Reviewer #2:

Hassan et al reported a point-of-care (PoC) device for potential sepsis management using neutrophils (nCD64) as a biomarker. The system was built upon some of the prior work from the same group but here for a different application. The system is interesting and the investigation included theoretical simulation, prototype development, and clinical study. However, the paper has several issues that need to be addressed/considered.

Our Response:

We appreciate the reviewer's comments that our paper is well demonstrated including theoretical simulation, biochip development and the clinical studies. Now, as you will see from this response, we have further enhanced the paper based on the reviewer's comments and added additional characterizations, results and discussion that provide further clarity and depth to our paper. The comments of the reviewer have made the paper better.

1) Patient characteristics including age, gender, type of sepsis/infection, etc. should be presented and discuss on whether/how these characteristics might affect the nCD64 being a biomarker for sepsis should be included.

Our Response:

We thank the reviewer for this comment. We have included a Supplementary Table providing more information on patient characteristics (including age, gender, type of infection etc.) for the data that we presented in Figure 2c.

Furthermore, nCD64 have also been reported to distinguish between bacterial vs. viral infections, with higher changes in expression in patients with bacterial infections [Ref. 47]. Another study have reported to combine nCD64 and expressions CD35 biomarkers and were able to distinguish between bacterial, viral and other inflammatory diseases [Ref. 48]. Furthermore, changes in the expression level is mostly age independent. There are numerous studies that have been conducted investigating the utility of nCD64 as biomarker for sepsis detection in neonates and infants. A meta-analysis Refs. 8-10] investigating nCD64 utility as a biomarker have shown the sepsis detection cut-off values of nCD64 to be independent of neonates, infants and adults. The increase/ decrease in expression can be related to microbe or severity of infection and not to age or gender.

We have added the above discussion in manuscript on page 16 line 10.

2) In Fig 2c, the authors reported the system improved AUC from .70 to .77; Please comment on how significant clinically this might be. Overall, while the general premise of this PoC system for more effective sepsis stratification is recognized, it is less clear how exactly this will improve clinical outcome. Does the new system improve any of these: accuracy, sensitivity, and specificity? And can it translate to improved patient survival, saved cost, etc. A discuss would be useful here.

Our Response:

We thank the reviewer for this comment. The improvement from 0.70 to 0.77 is statistically significant as also indicated by the p-value (< 0.001) reported in the Methods section. As shown in literature most of the studies were done in very restricted settings like ICU, where the patients have already confirmed bacterial infections, etc. In our study, we started with a heterogeneous patient population having several infections, and were able to see the statistical improvement in

prediction of sepsis. Delay in the early detection of sepsis can have multiple drawbacks. With the suspicion of infection, physicians usually recommend broad-range antibiotic treatment. Furthermore, inappropriate use of antibiotics, if sepsis and/ or infection is not diagnosed is harmful for the patients as it will interfere with the patient's normal microbiota, and potentially develop antimicrobial resistance to drugs.

An early sepsis stratification system could lead to improved screening techniques and more therapeutic options and potentially drastically reduce lengthy stays in critical care units, priced at >\$20,000 per stay on average for septic patients in the U.S. Furthermore, as the 'Pay for Performance' initiative is implemented in hospitals across the United States, their reimbursement is based on showing steadily improvement over time in critical statistics including lengths of stay, mortality rates, rates of hospital-acquired infections, and many others. And Sepsis has been identified as a Core Measure because better adherence to standardized sepsis guidelines has been shown to drastically improve all of these critical parameters. Therefore, we believe a more specific and sensitive method to perform accurate stratification of sepsis will not only improve patient outcomes but will be a critical part of the financial survival of hospitals across the United States.

We have added the above discussion in the manuscript at page 17 line 3-24.

3) Along the same line, especially, since different iterations of flow cytometry has been developed in the past; if only a minute of blood sample is required for each test, cannot we just dilute the blood with buffer and run through a portable flow cytometry without any "manual processing"? If flow cytometry profiling nCD64 has not made an impact in sepsis stratification, why would we expect this system in the paper to do so? Apart from being "complicated" for flow cytometry (which is debatable), are there any other hurdles that might prevent the presented system from making an immediate clinical impact?

Our Response: We appreciate the reviewers comment and would like to clarify further. nCD64 expression quantification on any flow cytometer will require the "manual processing" which includes (RBC lysis, conjugation with fluorescent antibodies and incubation in dark). Several benchtop flow cytometers with reduced complexity are currently being sold by companies like CytoBuoy and Cronus Technologies, as well as by larger companies such as Sysmex and Abbott. All of the above mentioned devices still require traditional flow cytometry sample preparation to be performed in a microbiology lab before the sample can be introduced to the device.

Furthermore, we would also like to add that just diluting the whole blood with buffer e.g. with 1x PBS will not help, since the ratio of WBC: RBC in whole blood is 1:500-1000, and for sensitive & accurate differentiation of leukocytes we need to lyse RBCs. This ratio would further reduce if we consider any specific type of leukocytes like nCD64 cells etc.

We believe a quick, POC testing (different settings like bed-side, or Emergency for trauma and/ or injured patients) (using flow cytometer or not), is the key here. nCD64 is a great pro-inflammatory biomarker at the onset of infection. The current blood based diagnostic process flow in the hospitals is prolonged: physician places a blood test order, phlebotomist comes to the patient room, draws blood and send it to flow lab for later processing. And during this time, the diagnostic window of opportunity can potentially be missed. Here, we believe our POC

system can have a clinical impact that it will bypass the entire current process flow and provide the testing with immediate availability of results within a matter of minutes to the physicians.

We thank the reviewer for this critical comment and have added some of the above discussion in manuscript at page 18 line 3-14.

4) "The 72 hour survival rate decreases by 7.7% every hour that appropriate antimicrobial medication is delayed," is misleading, as this only applies to the first few hours at onset of the condition.

Our Response:

We thank and agree with the reviewer. We have corrected this information in manuscript at page 3 line 23.

5) Is a drop of blood enough for accurate blood biomarker in this case? There has been recent evidence showing small volume blood (10 ul used in the study) could create more variation in blood marker detection. That is why current clinical testing typically uses a larger volume of blood.

Our Response:

We thank the reviewer for this comment and would like to clarify further. The volume of the blood (sample) required for the biomarker detection depends on the concentration of the biomarker in the blood. We believe 10uL of blood have enough leukocytes (dynamic range of concentration = 530-38570 cells/uL) for the sensitive and accurate detection and counting of cells. For example, 10uL volume of blood will definitely not be enough in the following cases where the biomarker concentration is very low:

- 1. Circulating Tumor Cell (CTC concentration: 1-10 cells/ mL) measurements from whole blood or*
- 2. Other specific leukocyte biomarkers e.g CD25-CD134+ T-cells (concentration: 1-5 cells/uL), potential biomarkers for latent TB detection.*

Furthermore, we agree with the reviewer that some of the recent studies have shown high variability in biomarker quantification from whole blood when collected from finger pricks (as shown below). However, we have used a drop of blood collected from venipuncture, which is easier to collect from the line port already installed in patient's arms. The need for using a small volume of blood although its availability is not an issue in this setting is still apparent in a POC device, to reduce the number of reagents, time and associated costs for a test.

We have added a Section, "Required Blood Volume" in Supplementary Information on page 3.

6) "The CD64+ cells are captured based on the CD64 expression level on their surfaces."; In order to support this statement, which is key for the study, the authors should provide more extensive theoretical and actual data to support this for a broader range of CD64 expression levels (not just high and low).

Our Response:

We thank the reviewer for the comment. Please see below our detailed response:

Flow Cytometry Data: Earlier, we have provided flow cytometry measurements for representative high, medium and low samples and their corresponding results in a blocked and antibody chamber (Supplementary Figures 16-18). Now, we have added Supplementary Figure 20 showing a diverse range of expression based cell capture results (experiments done on flow cytometer). The experimental protocol, cell capture vs. CD64 expression are shown in Supplementary Figure 20. We have also added the relevant information in manuscript on page 11 line 20.

Biochip (Electrical) Data: Current, manuscript Figure 3h shows cell capture results for a broad range of CD64 expression values from our biochip using $n = 102$ blood samples.

Theoretical Model: Now we have added a theoretical model outlining the direct relationship of increase in antigen density (expression level) on the cell's surface with increased probability of cell capture in Supplementary Information on page 1. We have also referenced this in manuscript on page 11 line 29.

Minor comments:

7) The list of references is not comprehensive enough, e.g. the following key reference should be cited: J. Cid et al. 2010. Neutrophil CD64 expression as marker of bacterial infection: A systematic review and meta-analysis. J. Infection. 60: 313–319.

Our Response:

We thank the reviewer for providing us with this reference. Now, you can see we have added this and more literature to provide a more comprehensive list of references for the readers. Newly added references are 8-10, 33-35, 38-48, 53 and 57.

8) In Figure 1, lysis buffer was used to break the red blood cells. Is the cell lysis mixture still in the channel? If the debris with comparable size passed through the electrodes and creates impedance changes, how could the system filter out those noises?

Our Response:

Yes, the lysis buffer is still in the channel along with the subsequently added quenching buffer. Initially, we add lysis buffer to lyse RBCs which changes the osmolarity/pH of the solution, we further add the quenching buffer to stop the lysing process and maintain osmolarity/ pH such that it will preserve the remaining leukocytes.

Yes, the RBC lyse and their debris tend to clump, the concentration of Saponin (which is known to dissociate debris clumps) in the lysis buffer is adjusted to effectively dissociate debris clumps. However, the dissociated micro debris particles pass through the counter and creates impedance changes. However, this impedance change (aka noise) is very small relative to the impedance change because of cells. We have also added the Supplementary Figure 24 showing in more detail the signal obtained because of a cell and relative signal associated with micro debris particle. We have also added the above information in manuscript on page 20 line 31.

9). In Figure 1b, how the pulse width is measured?

Our Response:

We have added another Supplementary Figure 25 to explain this for easier understanding of the readers. The pulse width is measured by finding number of samples in a positive pulse envelope above the detection threshold amplitude multiplied by sampling time as shown in the Figure. The above information is also added in methods section in manuscript at page 21 line 2.

10). In Figure 4, the flow rate was very briefly investigated. Capability to handle a broad range of flow rate is important to evaluate the sensitivity of the impedance sensing module and the throughput. A more thorough study on flow rate is highly recommended.

Our Response:

We thank the reviewers' comment and provide more information regarding flow rate optimization in both cases:

Sensitivity of the impedance Sensing Module:

In this study we varied the flow rate (10-30L/min) and investigated the noise levels in electrical signals, thereby impacting the sensitivity of the sensing module (electrical counter). We found that with increasing flow rates, the standard deviation of the baseline signal i.e. noise increases with increasing the flow rate. Supplementary Figure 9a shows that the increase in noise level with increase in flow rate and corresponding decrease in signal-to-noise ratios. Although the noise level is smallest at 10L/min, the increase in the noise at 30L/min, is still workable as the pulse peak amplitudes will still be detected. Another consideration will be the sampling rate, as increasing the flow rates will require increasing the sampling rate as well. Supplementary Figure 9b shows the average pulse width of the lymphocytes and granulocytes + monocytes population, which decreases with increasing flow rates. Furthermore, the required sampling rate to keep same sampling resolution increases linearly. Currently, we have sampled the signal at 250 kHz, which provides enough resolution for pulse detection (each pulse comprised of 75-100 data points). To keep the same sampling resolution of the each electrical pulse at higher flow rates, the sampling rate should be increased linearly with the flow rate. We have added the corresponding Supplementary Figure 9.

Increased Throughput vs. Flow Rate:

In case of the throughput optimization, as the flow rate will be increased the experimental assay time will be reduced. Currently at 10ul/min, it takes around 25 min to finish the complete assay. However, simple device modifications can easily result in much reduced experimental time of ~10 min. Some of the necessary modifications include:

- 1. In order to keep the lysing/ quenching times constant for optimized lysing process at higher flow rates, we need to increase the volume of the lysing chip, which can be easily done by increasing the height of the lysing and quenching channels.*
- 2. Similarly, increasing the flow rate for higher throughput can change the shear stress in the chamber, thus producing different conditions for cell capture. To obtain the same shear stress, a simple modification is to increase the height of the chamber to keep the similar capture conditions for CD64 cell capture.*

Supplementary Table 4 shows the flow rate characterization with respect to the biosensor throughput. It provides information on the necessary dimensional changes required in the biosensor design with increasing flow rates, which will also result in the corresponding decrease in the experimental assay time

Furthermore, we also varied and tested the flow rates to show that non-specific capture in the blocked chamber is minimum as shown by current Supplementary Figure 19.

Now, we have provided the above detailed information on flow rate characterization in Supplementary Information on page 2. We have also briefly referenced it in manuscript on page 18 line 15.

11). Supplementary material, Figure 1, it seems the connection pads are connected to traces on PCB using soldering paste. The traces are long and the detection wires seem not protected by any shield. Would the sensitivity of Impedance changes by micro level particles be compromised by this system setup?

Our Response:

In this extensive study we had various quality control procedures including one for the counter. We tested each counter before using in the experiment. Test of the counter includes testing for any shorts after applying the conductive epoxy and also measuring resistance from the electrode to the PCB lines. As a threshold we used <5 Ohms resistance in between the electrode and PCB connecting lines. Now, we have discussed this in manuscript too on page 19 line 31.

12). Supplementary material, Figure 5, why the x-axis is not time?

Our Response:

In Supplementary Figure 5, the x-axis is the number of samples. Now, we have changed the x-axis to time instead of number of samples in the updated Figure.

13) Fig 2 d and e, and Fig S8 and S9, what are those red color cross bars?

Our Response:

Thank you for the comment. The red color crosses are outliers. We like to provide further clarification. We have used Matlab function “boxplot” to plot those figures. The “boxplot” function of Matlab draws points as outliers if they are greater than $q_3 + 1.5 \times (q_3 - q_1)$ or less than $q_1 - 1.5 \times (q_3 - q_1)$. q_1 and q_3 are the 25th and 75th percentiles of the sample data. We have added this information in methods section in the manuscript on page 24 line 30 for easier understanding of the readers.

14) There are several typos and grammar issues in the text. The authors should carefully proof-read the manuscript.

Our Response:

Thank you for pointing this out. We have proof-read the manuscript for typos and errors.

15) I am also not sure if the clinical studies performed in the study can be considered as “clinical trials”.

Our Response:

We thank and agree with the reviewers comment. We also after further discussed with the clinicians and agreed that this was not an interventional clinical study (aka clinical trial with decision making for patients based on a test), and should be considered as a “clinical study”. We have revised a manuscript accordingly.

REVIEWERS' COMMENTS:

Reviewer #1 (Remarks to the Author):

The authors adequately addressed the major issues as well as made multiple changes to improve the format of the manuscript to make it easier to follow and read.

Reviewer #2 (Remarks to the Author):

my comments have been properly addressed. The revised manuscript has been improved overall.